# Pooled genome-wide CRISPR activation screening for rapamycin resistance genes in *Drosophila* cells

**Baolong Xia[1], Raghuvir Viswanatha[1], Yanhui Hu[1,2], Stephanie E Mohr[1,2], Norbert Perrimon[1,2,3]***

[1]Department of Genetics, Blavatnik Institute, Harvard Medical School, Boston, United States; [2]Drosophila RNAi Screening Center, Harvard Medical School, Boston, United States; [3]Howard Hughes Medical Institute, Boston, United States

**Abstract** Loss-of-function and gain-of-function genetic perturbations provide valuable insights into gene function. In *Drosophila* cells, while genome-wide loss-of-function screens have been extensively used to reveal mechanisms of a variety of biological processes, approaches for performing genome-wide gain-of-function screens are still lacking. Here, we describe a pooled CRISPR activation (CRISPRa) screening platform in *Drosophila* cells and apply this method to both focused and genome-wide screens to identify rapamycin resistance genes. The screens identified three genes as novel rapamycin resistance genes: a member of the SLC16 family of monocarboxylate transporters (*CG8468*), a member of the lipocalin protein family (*CG5399*), and a zinc finger C2H2 transcription factor (*CG9932*). Mechanistically, we demonstrate that *CG5399* overexpression activates the RTK-Akt-mTOR signaling pathway and that activation of insulin receptor (InR) by *CG5399* requires cholesterol and clathrin-coated pits at the cell membrane. This study establishes a novel platform for functional genetic studies in *Drosophila* cells.

## Editor's evaluation

This work is well-structured, with clear objectives and experiments. The authors successfully demonstrated genome-wide gene activation which overcame previous failed attempts to replicate gene activation that worked well in mammalian systems. The study is detailed and relevant for the application of CRISPRa in understanding the function of candidate genes.

**\*For correspondence:**
perrimon@genetics.med.
harvard.edu

**Competing interest:** The authors declare that no competing interests exist.

## Introduction

Although *Drosophila* is one of the most intensively studied organisms, about half of *Drosophila* protein-coding genes still lack functional characterization (*Ewen-Campen et al., 2017*). Genome-scale loss-of-function (LOF) and gain-of-function (GOF) genetic perturbations facilitate functional genomic studies. In *Drosophila*, genome-wide LOF screens using RNA interference or CRISPR-Cas9 have helped elucidate the mechanisms of a variety of biological processes (*Boutros et al., 2004*; *Björklund et al., 2006*; *D'Ambrosio and Vale, 2010*; *Bard et al., 2006*; *Guo et al., 2008*; *Hao et al., 2008*; *Song et al., 2022*; *Viswanatha et al., 2018*). However, genome-wide GOF screens have not been feasible, largely due to a lack of the reagents for genome-scale GOF perturbation. Current state-of-the-art for *Drosophila* GOF studies utilizes cDNA library overexpression (*Stapleton et al., 2002*; *Yu et al., 2011*), which only covers a subset of the genome and underrepresents genes with long open reading frames (ORFs).

CRISPRa is a complementary approach for GOF studies. Nuclease-dead Cas9 (dCas9) is fused with different transcriptional activators and targeted to the promoter region by single-guide RNAs (sgRNAs) to facilitate transcription at endogenous loci. Synergistic activation mediator (SAM) complex is one of the most efficient CRISPRa systems (*Konermann et al., 2015*; *Chavez et al., 2016*). With the SAM system, synthetic transcriptional activators dCas9-VP64 and MCP-p65-HSF1 are recruited to the promoter region of endogenous genes by MS2 hairpin-containing sgRNAs. Genome-scale perturbation with CRISPRa can be achieved by synthesizing a genome-wide sgRNA library, bypassing the need to generate a genome-wide ORF library. CRISPR-based GOF approaches have been used for genome-wide genetic screens in mammalian cells (*Konermann et al., 2015*; *Rodríguez et al., 2022*; *Zhu et al., 2022*; *Sofer et al., 2022*; *Zhang et al., 2022*), but the feasibility has not yet been tested in *Drosophila* cells.

To address this gap, we developed a pooled genome-wide CRISPRa screening platform in *Drosophila* cells using the SAM complex. To demonstrate the feasibility of this platform, we performed focused and genome-wide CRISPRa screens and identified novel candidates conferring rapamycin resistance. Next, we focused on one of the top candidates, a member of the lipocalin protein family (*CG5399*), and demonstrate that it positively regulates the Receptor Tyrosine Kinase (RTK)-Akt-mTOR signaling pathway by regulating cholesterol and clathrin-coated pits at the cell membrane.

## Results

### Inducible CRISPRa using the SAM complex in *Drosophila* S2R+ cells

To establish CRISPRa in *Drosophila* cells in an inducible manner, the synthetic transcriptional activators of the SAM complex (dCas9-VP64 and MCP-p65-HSF1) were placed under a metallothionein promoter, which can be induced in the presence of copper ions. VP64, p65, and HSF1 are transcriptional activators while the MS2 coat protein (MCP) recognizes and binds to MS2 hairpins present in the sgRNAs, recruiting the fused transcriptional activators to gene promoters targeted by sgRNAs. In addition, the MS2 hairpin-containing sgRNA was expressed from a separate plasmid under the control of the U6 promoter (*Figure 1A*). To determine whether this system can mediate transcriptional activation in *Drosophila* cells, plasmids with sgRNAs targeting the promoter regions of *Jon25Biii* or *Sdr* were transfected into *Drosophila* S2R+ cells stably expressing the metallothionein promoter-driven SAM complex. Without copper induction, sgRNAs moderately activated the target genes, likely due to leaky expression of the SAM complex from the metallothionein promoter. In the presence of copper,

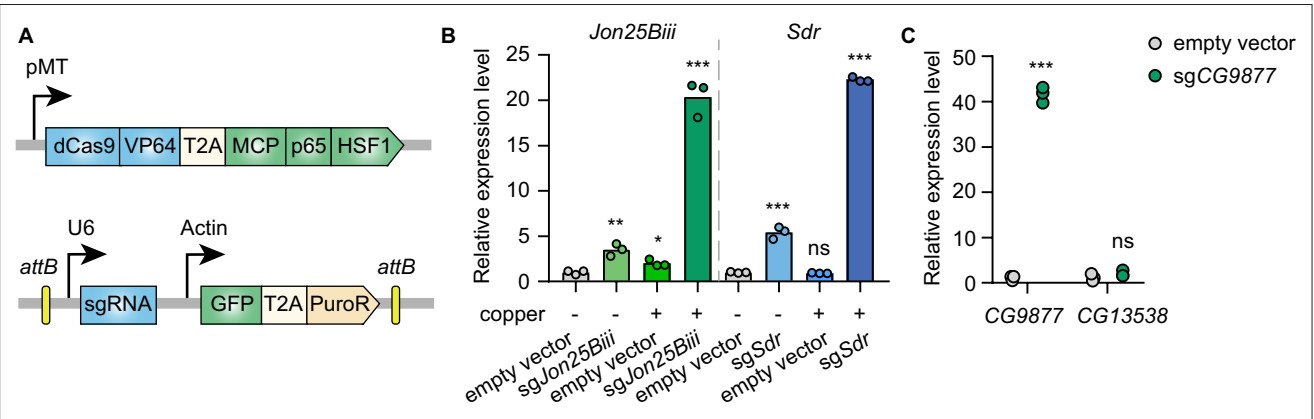

**Figure 1.** Inducible transcriptional activation by the synergistic activation mediator (SAM) complex in *Drosophila* cells. (**A**) Schematic of the SAM complex for inducible transcriptional activation. dCas9-VP64 and MCP-p65-HSF1 were driven by an inducible metallothionein promoter. dCas9-VP64 and MCP-p65-HSF1 were expressed as T2A-containing bicistronic transcript. single-guide RNA (sgRNA) was expressed from pLib8 plasmid, which contains an *attB* flanking GFP-T2A-PuroR cassette for *attP* sites recombination. (**B**) Fold activation of *Jon25Biii* and *Sdr* expression measured by qPCR. Three biological replicates are shown as individual circles. (**C**) Fold activation of *CG9877* and *CG13538* expression measured by qPCR. Three biological replicates are shown as individual circles. *t*-test, *p<0.05; **p<0.01; ***p<0.001; ns, not significant.

The online version of this article includes the following source data for figure 1:

**Source data 1.** Full source data for *Figure 1*.

sgRNAs robustly upregulated the target genes (*Figure 1B*), showing that the SAM complex is able to mediate gene activation in *Drosophila* S2*R*+ cells.

The SAM complex modulates gene expression by recruiting transcriptional activators to gene promoters at endogenous genomic loci. In the *Drosophila* genome, 32% of genes form divergent gene pairs with transcriptional start sites (TSS) <1 kb apart (*Yang and Yu, 2009*). Thus, we next assessed the potential for collateral activation by the SAM complex of the closely spaced promoters. *CG9877* and *CG13538* are a divergent gene pair with transcriptional start sites 908 bp apart, thus this gene pair is a good test case for assessing collateral activation. sgRNAs were designed within a region 300 bp upstream of the TSS of *CG9877* and ~600 bp away from the TSS of *CG13538*. The sgRNAs specifically activated *CG9877*, but not *CG13538* (*Figure 1C*), suggesting that the SAM complex activates the target gene without affecting nearby genes at least in some divergent gene pairs.

## Pooled CRISPRa screening with a focused library

Next, we used the SAM complex to perform a pooled CRISPRa screen. Unlike in mammalian cells, lentivirus vectors are extremely inefficient in *Drosophila* cells. To overcome this limitation, we previously established a pooled library delivery method based on site-specific recombination following plasmid transfection of *Drosophila* cells (*Viswanatha et al., 2018*). In this method, *attB* sites flanking sgRNAs are integrated into *attP* sites flanking landing cassette in the presence of phiC31 integrase. We used the S2*R*+ PT5 cell line in which the *attP* cassette is inserted into the *Clic* locus (*Neumüller et al., 2012*). To establish pooled library cells for CRISPRa screens, we first generated SAM cells that stably expressed the metallothionein promoter-driven SAM complex. Then, using phiC31-mediated cassette exchange, we integrated a pooled guide RNA library into the landing cassette (*Figure 2— figure supplement 1A*).

To test the pooled screen approach, we performed a CRISPRa screen to identify rapamycin resistance genes. Rapamycin is an allosteric inhibitor of the kinase mTOR, a key regulator of cell growth and proliferation. Rapamycin inhibits *Drosophila* S2*R*+ cell proliferation in a dose-dependent manner. Cell proliferation was partially inhibited by rapamycin at 0.1 nM and almost completely inhibited at 1 nM (*Figure 2—figure supplement 1B*). To identify a suitable rapamycin treatment condition for a pooled CRISPRa screen, we performed a pilot-focused screen with different rapamycin concentrations and treatment durations. The focused library consists of 6335 sgRNAs targeting the promoter regions of 652 genes, which include genes involved in the mTOR signaling pathway as well as candidates from our previous CRISPR knockout screen (*Viswanatha et al., 2018*). Pooled library cells were subjected to passaging in 0.1 nM, 1 nM rapamycin, or DMSO (vehicle) containing culture medium in parallel for 15 days or 30 days. After treatment, we examined sgRNA abundance in each condition by next-generation sequencing. First, we identified genes affecting cell fitness by comparing the final population after passaging in DMSO with the initial population. *Scyl* and *Cyp12a4* are significantly depleted (false-discovery rate (FDR)<0.05) after passaging in DMSO for 15 days and 30 days (*Figure 2—figure supplement 1C and D*), suggesting that overexpression of *scyl* or *Cyp12a4* affects cell fitness. Consistent with this, *scyl* is known to inhibit cell proliferation and act as a cell death activator (*Reiling and Hafen, 2004*; *Corradetti et al., 2005*; *Scuderi et al., 2006*). These results demonstrate that pooled CRISPRa screen can be used to identify genes affecting cell fitness.

We next identified rapamycin resistance genes by comparing the rapamycin-treated population with the DMSO-treated population, reasoning that sgRNAs that lead to activation of rapamycin resistance genes would be enriched after treatment due to a growth advantage in the presence of rapamycin. The focused screen revealed that one candidate, *CG8468*, was significantly enriched (FDR<0.05) in the population treated with 1 nM rapamycin. Moreover, with prolonged treatment, *CG8468* was further enriched as we observed a higher fold-change value at day 30 than at day 15 (*Figure 2—figure supplement 1E and F*). However, no gene was significantly enriched in the 0.1 nM rapamycin treatment condition, probably because cells have a higher proliferation rate at 0.1 nM rapamycin concentration compared to 1 nM. Taken together, these data demonstrate that pooled CRISPRa screen using the SAM complex is feasible in *Drosophila* cells.

## Genome-wide pooled CRISPRa screen

Next, we sought to screen for rapamycin resistance genes at the genome-wide scale. A previous study showed that multiplexed sgRNAs performed better than single sgRNAs for CRISPRi and CRISPRa

(*Replogle et al., 2020*). Thus, we designed a dual-sgRNA library in which each vector expressed two distinct sgRNAs targeting the promoter region of the same gene within 500 bp upstream of the TSS (*Figure 2—figure supplement 2A*). The library consists of 84,143 vectors targeting the promoter regions of 13,293 protein-coding genes and 2332 non-coding genes. The dual-sgRNA library was constructed using a three-step pooled cloning strategy (*Figure 2—figure supplement 2B* and Methods). After library construction, we checked the quality of the dual-sgRNA library by deep sequencing, which revealed that ~98.5% of designed vectors were present in the final library. The difference in representation between the 10th percentile of the final library (19 reads) and the 90th percentile (292 reads) was 15.5-fold (*Figure 2—figure supplement 2C*). The integrity and distribution of our dual-sgRNA library are comparable to other published genome-wide libraries (*Sanjana et al., 2014*), indicating the high quality of the library.

To establish the pooled library cells, we integrated the genome-wide dual-sgRNA library into SAM cells by phiC31-mediated cassette exchange. The pooled library cells were then passaged in 1 nM rapamycin or DMSO-containing medium for 3 weeks. The abundance of dual-sgRNA vectors in the initial and final cell populations following rapamycin or DMSO treatment was analyzed by next-generation sequencing (*Figure 2A*). First, we identified genes affecting cell fitness by comparing the final population after passaging in DMSO for 3 weeks with the initial population. Consistent with the results of the focused screen, *scyl* is also significantly depleted (FDR<0.05) in the genome-wide screen dataset. In addition to the *scyl*, the screen also identified other genes known to be involved in the suppression of cell proliferation (*Figure 2—figure supplement 3A* and *Table 1*).

By comparing the rapamycin-treated population with the DMSO-treated population, we identified rapamycin resistance genes. *Pka-C3* and *Cdc25* were among the top 50 ranked genes identified in both replicates of the genome-wide screen (*Table 2*). *Pka-C3* encodes the catalytic subunit of PKA, and overexpression of the catalytic subunit of PKA or activation of the PKA pathway is known to confer resistance to rapamycin (*Cutler et al., 2001*; *Zurita-Martinez and Cardenas, 2005*; *Schmelzle et al., 2004*). *Cdc25* is a tyrosine phosphatase gene that regulates cell cycle progression. Previous studies have indicated that the level of *Cdc25* expression is positively correlated with rapamycin resistance in cancer cells (*Reikvam et al., 2014*; *Chen et al., 2009*).

In addition to the known rapamycin resistance genes, novel candidates were also identified in the CRISPRa screen. In particular, three genes, *CG8468*, *CG5399*, and *CG9932*, were significantly enriched (FDR<0.05) in both replicates (*Figure 2B*). *CG8468* is a member of the SLC16 family of monocarboxylate transporters. *CG5399* encodes a member of the lipocalin protein family, members of which have been implicated in lipid binding and transport. *CG9932* encodes a zinc finger C2H2 transcription factor. Interestingly, *CG8468* was also the top hit from our focused library screen, indicating the consistency of the pooled CRISPRa screen approach in *Drosophila* cells (note that the other two genes were not included in the focused library).

## Validation of the novel rapamycin resistance genes

To validate that the novel hits from the CRISPRa screen could indeed confer resistance to rapamycin, we first cloned all dual-sgRNA vectors targeting *CG8468*, *CG5399*, or *CG9932* that were present in the genome-wide library and established individual stable SAM cell lines for each vector. The target activation efficiency of each vector was evaluated in individual cell lines by qPCR. The dual-sgRNA vectors showed variable activation efficiency, probably due to the complex transcriptional regulation of the target genes or different sgRNA binding efficiencies (*Figure 2C, F and I*). Interestingly, the enrichment of each dual-sgRNA vector in the screen was highly correlated with its target activation efficiency, as only the vectors that efficiently activate target genes were enriched in the rapamycin-treated samples (*Figure 2D, G and J* and *Figure 2—figure supplement 3B–D*). Collectively, these results confirm that the sgRNA vectors enriched in the screen were able to upregulate the target genes.

To validate that overexpression of the hits confers a growth advantage in the presence of rapamycin, we mixed wild-type SAM cells (GFP negative) and individual dual-sgRNA vector expressing cell lines (GFP positive), then monitored the proportion of GFP-positive cells in the mixed cell populations following 1 nM rapamycin or DMSO treatment for 2 weeks. We reasoned that if a dual-sgRNA vector confers resistance to rapamycin, cells with the vector will proliferate more than wild-type SAM cells in the presence of rapamycin, leading to a higher proportion of GFP-positive cells in the

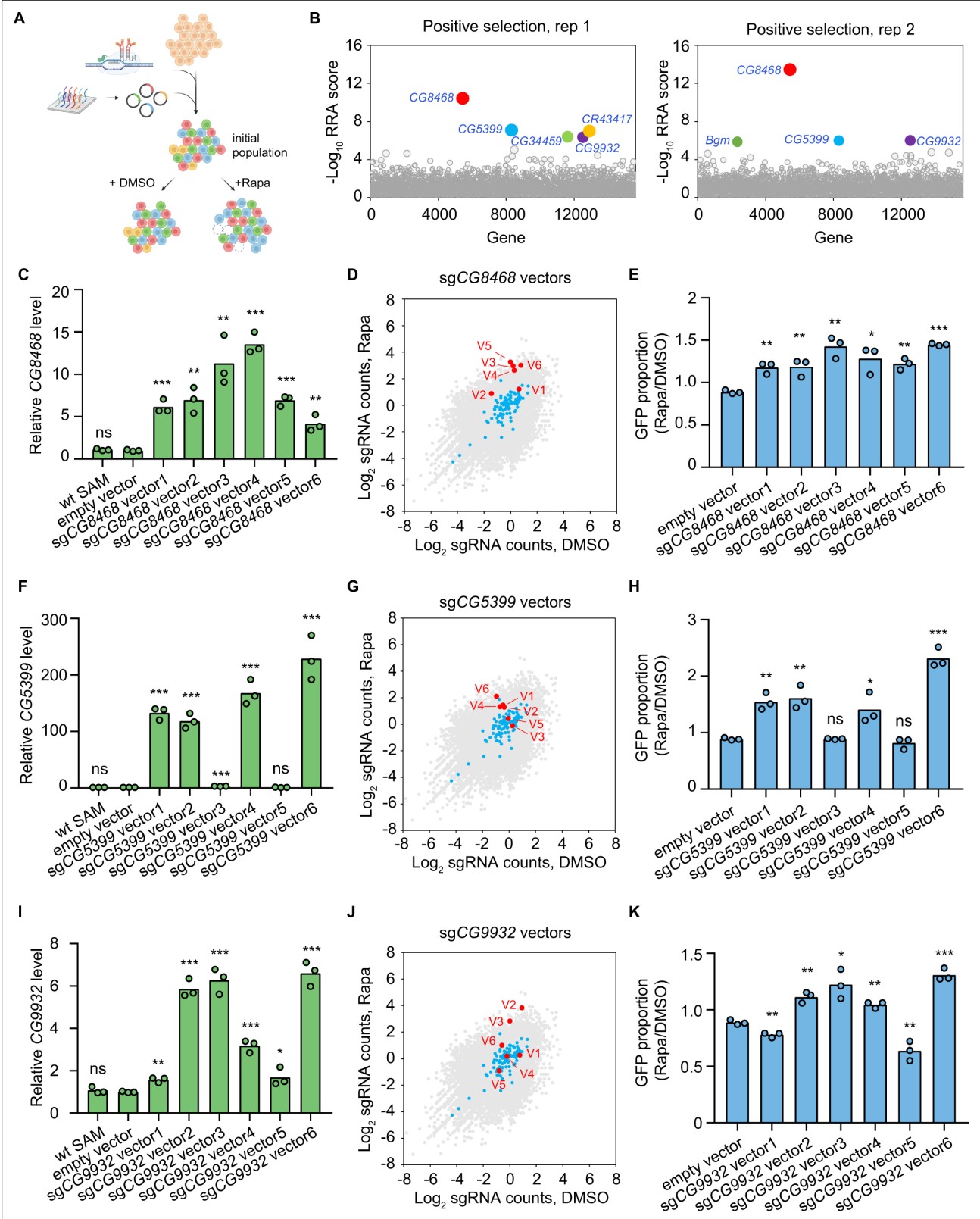

**Figure 2.** Genome-wide CRISPR activation screen for rapamycin resistance genes. (**A**) Schematic of CRISPR activation screen (See methods). (**B**) Two replicates of genome-wide CRISPR activation screen. Data were analyzed by MAGeCK-RRA, a smaller RRA score indicates a stronger selection effect. Each circle represents a gene. Circle size corresponds to the significance (p value) of enrichment. Significantly enriched genes (false-discovery rate (FDR)<0.05) are colored. (**C**) Fold activation of *CG8468* expression measured by qPCR. Three biological replicates are shown as individual circles.

*Figure 2 continued on next page*

*Figure 2 continued*

(**D**) Counts of sg*CG8468* vectors from the genome-wide screen. Each dot represents a vector. Vectors targeting intergenic regions are shown in blue. Vectors targeting *CG8468* are shown in red and annotated as V1-V6. (**E**) sg*CG8468*-expressing cell proliferation in cell mixture following 1 nM rapamycin or DMSO treatment. GFP proportion was measured by flow cytometry. Three biological replicates are shown as individual circles. (**F**) Fold activation of *CG5399* expression measured by qPCR. Three biological replicates are shown as individual circles. (**G**) Counts of sg*CG5399* vectors from the genome-wide screen. Each dot represents a vector. Vectors targeting intergenic regions are shown in blue. Vectors targeting *CG5399* are shown in red and annotated as V1-V6. (**H**) sg*CG5399*-expressing cell proliferation in cell mixture following 1 nM rapamycin or DMSO treatment. GFP proportion was measured by flow cytometry. Three biological replicates are shown as individual circles. (**I**) Fold activation of *CG9932* expression measured by qPCR. Three biological replicates are shown as individual circles. (**J**) Counts of sg*CG9932* vectors from the genome-wide screen. Each dot represents a vector. Vectors targeting intergenic regions are shown in blue. Vectors targeting *CG9932* are shown in red and annotated as V1-V6. (**K**) sg*CG9932*-expressing cell proliferation in cell mixture following 1 nM rapamycin or DMSO treatment. GFP proportion was measured by flow cytometry. Three biological replicates are shown as individual circles. *t*-test, *p<0.05; **p<0.01; ***p<0.001; ns, not significant.

The online version of this article includes the following source data and figure supplement(s) for figure 2:

**Source data 1.** Full source data for *Figure 2*.

**Figure supplement 1.** Pooled CRISPR activation screen with a focused library.

**Figure supplement 1—source data 1.** Full source data for *Figure 2—figure supplement 1*.

**Figure supplement 2.** Design of genome-wide dual-sgRNA library.

**Figure supplement 2—source data 1.** Full source data for *Figure 2—figure supplement 2*.

**Figure supplement 3.** Genome-wide cell fitness screen and rapamycin screen.

**Figure supplement 3—source data 1.** Full source data for *Figure 2—figure supplement 3*.

rapamycin-treated sample as compared to the DMSO-treated sample. As expected, we only observed higher proportions of GFP for vectors that efficiently activate target genes, but not for inefficient vectors or an empty vector (*Figure 2E, H and K*). These results confirmed that overexpression of the hits from the genome-wide screen conferred resistance to rapamycin.

**Table 1.** Significantly depleted genes in genome-wide fitness screen.

| Gene | Human ortholog | Known gene affecting cell fitness | Reference |
|---|---|---|---|
| zld | ZNF485 | | |
| Eaat1 | SLC1A3 | | |
| CR44587 | - | | |
| Lis-1/Ptp52F* | PAFAH1B1/Ptprb | LIS-1-overexpressing mitotic cells show a variety of spindle defects | PMID: 10722879 |
| scyl | DDIT4 | Scyl inhibits cell growth by regulating the Tor pathway | PMID: 15545626 |
| αTub85E | TUBA1A | | |
| Poxm | Pax9 | PAX9 overexpression inhibits cancer cell proliferation | PMID: 35628401 |
| Dll | DLX6 | | |
| LKRSDH | AASS | Overexpression of Aass suppresses cancer cell proliferation | PMID: 31601242 |
| scro | NKX2-1 | NKX2-1 suppresses lung cancer progression by dampening ERK activity | PMID: 34689179 |
| CG3168 | SV2A | Overexpression of SV2A inhibits the PI3K signaling pathway | PMID: 34277597 |
| CG2930 | SLC15A1 | | |

*Lis-1 and Ptp52F form divergent gene pair ~500 bp apart.

**Table 2.** Overlapping genes of top-ranked 50 hits from two genome-wide screen replicates.

| Rank (Rep 1, Rep 2) | Gene | Human ortholog | Function | Known rapamycin resistance gene |
|---|---|---|---|---|
| 1, 1 | CG8468 | SLC16A8 | monocarboxylate transporter | |
| 2, 3 | CG5399 | APOD/LCN2 | lipocalin | |
| 5, 2 | CG9932 | ZFN462/REST | transcription factor | |
| 4, 10 | CG34459 | / | unknown | |
| 22, 13 | Pka-C3 | PRKX | catalytic subunit of PKA | PMID: 15643061, 14673167, 11739804 |
| 41, 8 | Ps | NOVA1 | RNA splicing | |
| 43, 45 | CDC25 | CDC25A/CDC25B | tyrosine phosphatase | PMID: 24383842, 19276368 |

## RTK-Akt-mTOR signaling activation by *CG5399* overexpression

To characterize the mechanism of rapamycin resistance, we first examined the mTOR activity in SAM cell lines with dual-sgRNA vectors activating the target genes. As ribosomal protein S6 is phosphorylated by S6K, which is a downstream target of mTOR, the phosphorylation status of S6 can serve as a readout for mTOR activity. In the presence of rapamycin, S6 phosphorylation was strongly inhibited in wild-type SAM cells and empty vector-expressing SAM cells. In contrast, compared to control cells, the phospho-S6 levels were dramatically elevated in *CG5399*-overexpressing cells (*Figure 3A*). Interestingly, we did not observe higher phospho-S6 levels in *CG8468*-overexpressing cells, suggesting that *CG8468* acts downstream of mTOR or in a parallel pathway. *CG9932*-overexpressing cells also displayed higher phospho-S6 levels, possibly reflecting a previous observation that overexpression of the human ortholog of *CG9932*, REST, activates Akt, which acts upstream of S6 (*Dobson et al., 2019*).

The increase of phospho-S6 levels in *CG5399*-overexpressing cells in the presence of rapamycin might be explained by different mechanisms: (1) *CG5399* might alter the pharmacokinetics of rapamycin, decreasing cellular rapamycin concentrations; (2) *CG5399* might competitively bind to rapamycin, releasing mTOR from inhibition; (3) *CG5399* might be a positive regulator of mTOR signaling. To distinguish among these possibilities, phospho-S6 was assessed in *CG5399*-overexpressing cells without rapamycin treatment. Compared with wild-type SAM cells and empty vector-expressing SAM cells, higher phospho-S6 levels were observed in *CG5399*-overexpressing cells (*Figure 3B*), indicating that *CG5399* is a positive regulator of mTOR. Moreover, in addition to higher phopho-S6 levels, an increase in phospho-Akt was also observed in *CG5399*-overexpressing cells. Furthermore, knocking down *CG5399* mRNA levels using either of two nonoverlapping double-stranded RNAs (dsRNAs) in *CG5399*-overexpressing cells totally abolished the increase of phospho-Akt and phospho-S6 (*Figure 3C*), excluding the possibility that an off-target effect of the sgRNAs explains these observations.

Akt is phosphorylated by PI3K when receptor tyrosine kinases (RTKs) are activated. To test whether PI3K is involved in Akt activation by *CG5399* overexpression, we used two nonoverlapping dsRNAs to deplete the catalytic subunit of PI3K, *Pi3K92E*, in *CG5399*-overexpressing cells. Knockdown of *Pi3K92E* abolished Akt activation by *CG5399* overexpression, suggesting that *CG5399* activates Akt-mTOR through PI3K (*Figure 3—figure supplement 1A*). As Akt is regulated by both insulin receptor (InR) and PDGF/VEGF receptor (Pvr) (*Sopko et al., 2015*), we examined whether *InR* and *Pvr* are involved in *CG5399* function. Two nonoverlapping dsRNAs targeting *InR* or *Pvr* were transfected into *CG5399*-overexpressing cells. Knockdown of *InR* or *Pvr* inhibited upregulation of phospho-Akt and phospho-S6 induced by *CG5399* overexpression (*Figure 3D*, *Figure 3—figure supplement 1B*), suggesting that *CG5399* activates Akt-mTOR via *InR* and *Pvr*. As expected, higher phospho-InR levels were also observed in *CG5399*-overexpressing cells (*Figure 3E*) in a normal medium without insulin stimulation. Finally, the activation of InR-Akt-mTOR signaling in S2R+ cells was also observed with *CG5399* ORF overexpression by co-transfecting *pUAS-CG5399* and *pActin-Gal4* vectors (*Figure 3F*), further excluding the possibility that off-targets of sgRNAs contribute to the phenotype. Taken

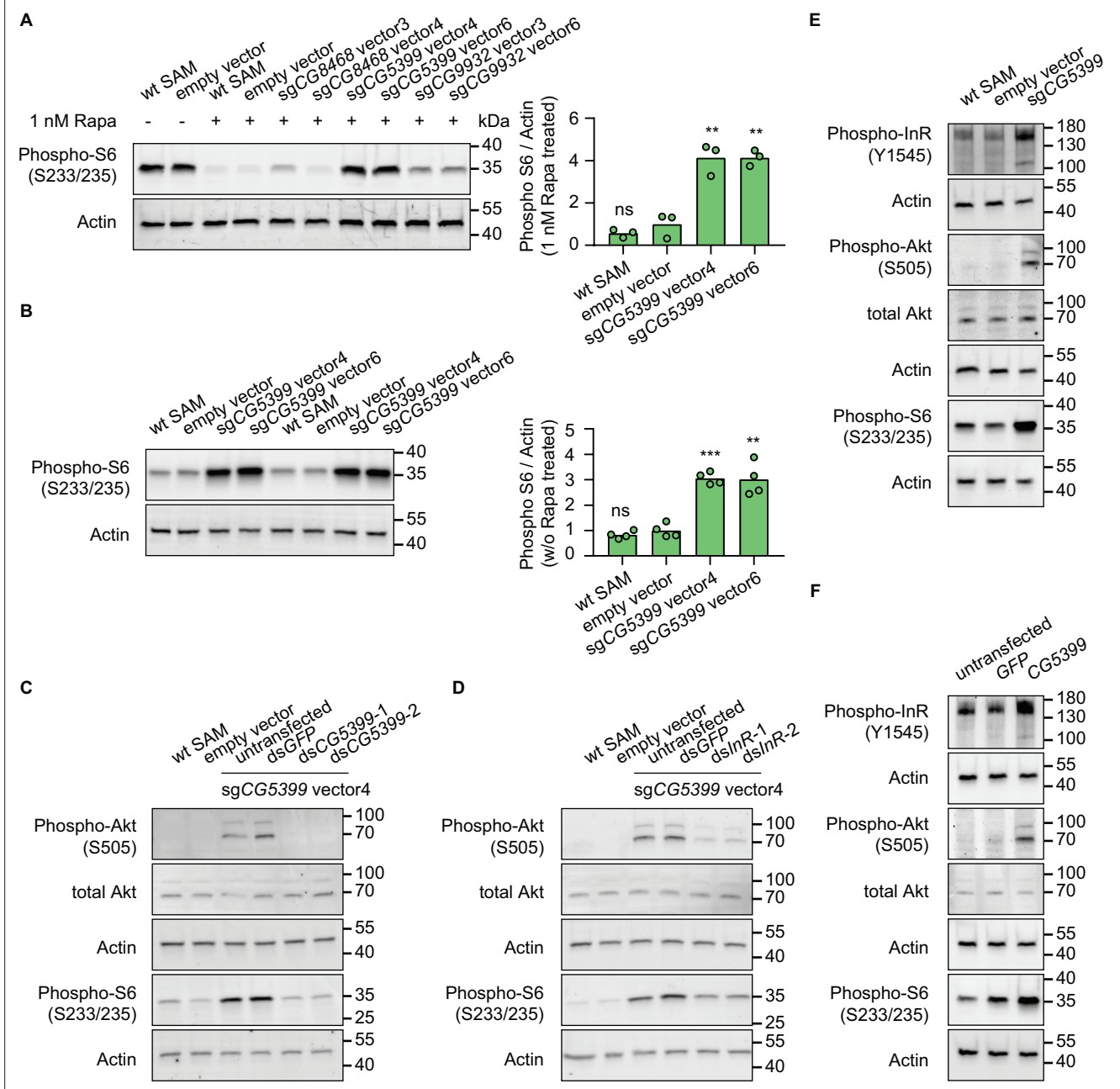

**Figure 3.** *CG5399* overexpression activates RTK-Akt-mTOR signaling. (**A**) Phospho-S6 levels in cells expressing dual-sgRNA vectors in the presence of 1 nM rapamycin. Western blot signals are quantitatively analyzed by ImageJ. Three biological replicates are shown as individual circles. (**B**) Phospho-S6 levels in cells expressing sg*CG5399* vectors without rapamycin treatment. Western blot signals are quantitatively analyzed by ImageJ. Four biological replicates are shown as individual circles. (**C**) Phospho-Akt and Phospho-S6 in *CG5399*-overexpressing cells following *CG5399* knockdown. Two nonoverlapping double-stranded RNAs (dsRNAs) targeting *CG5399* were used. (**D**) Phospho-Akt and Phospho-S6 in *CG5399*-overexpressing cells following insulin receptor (*InR*) knockdown. Two nonoverlapping dsRNAs targeting *InR* were used. (**E**) Phospho-InR, phospho-Akt, and phospho-S6 in sg*CG5399*-expressing synergistic activation mediator (SAM) cells. (**F**) Phospho-InR, phospho-Akt, and phospho-S6 in *CG5399* ORF-overexpressing S2R+ cells using UAS-Gal4. *t*-test, **<0.01; ***p<0.001; ns, not significant.

The online version of this article includes the following source data and figure supplement(s) for figure 3:

Source data 1. Full source data for *Figure 3*.

Figure supplement 1. *CG5399* overexpression activates Akt-mTOR through RTK/PI3K.

Figure supplement 1—source data 1. Full source data for *Figure 3—figure supplement 1*.

together, these data demonstrate that *CG5399* overexpression activates RTK-Akt-mTOR signaling in a normal medium without insulin stimulation.

Insulin receptors form covalent homodimers at the cell surface. Upon insulin binding, the ectodomain of InR changes from the inverted U-shape structure to the T-shape structure, facilitating the proximity and autophosphorylation of the cytoplasmic kinase domains (*Gutmann et al., 2018*; *Scapin et al., 2018*). *Drosophila* cells are cultured in Schneider medium supplemented with 10% FBS (fetal bovine serum). The Schneider medium only consists of amino acids and inorganic salts while FBS is a biological product that might contain a trace amount of insulin and insulin-like growth factors (*Tu et al., 2018*). As we observed that *CG5399* overexpression activates the InR in a normal medium without insulin stimulation, we removed FBS from the culture medium to further exclude the effect of insulin and insulin-like growth factors in FBS. Increase of phospho-InR and phospho-Akt in *CG5399*-overexpressing cells could still be observed after serum starvation for 2 hr (*Figure 3—figure supplement 1C*), suggesting InR activation by *CG5399* overexpression does not require insulin.

## InR regulation by *CG5399* requires cholesterol and clathrin-coated pits

InR is embedded in the lipid bilayer environment of the cell membrane. Given that *CG5399* encodes a member of the lipocalin protein family, members of which have been implicated in the binding and transport of lipid molecules, we hypothesized that *CG5399* might regulate InR by remodeling lipid components at the cell membrane. Structure prediction of CG5399 by AlphaFold revealed a highly conserved barrel structure formed by eight beta-sheets as a putative ligand pocket (*Figure 4—figure supplement 1A*), similar to the crystal structures of lipocalins in other species (*Breustedt et al., 2005*; *Lakshmi et al., 2015*). Molecular docking simulation indicated that cholesterol can be inserted into the barrel structure of CG5399 (*Figure 4—figure supplement 1B*), suggesting that cholesterol might be a substrate of CG5399.

To test whether cholesterol is relevant to *CG5399* function, we used methyl-beta-cyclodextrin (MβCD) to deplete cholesterol from cell membranes in *CG5399*-overexpressing cells. MβCD is a heptasaccharide with a high affinity to cholesterol and has been widely used to manipulate membrane cholesterol content (*Zidovetzki and Levitan, 2007*). MβCD treatment eliminated the increase of phospho-InR, phospho-Akt, and phospho-S6 in a dose-dependent manner in *CG5399*-overexpressing cells (*Figure 4A*), indicating that activation of InR-Akt-mTOR by *CG5399* overexpression requires cholesterol at the membrane. Moreover, supplementation of cholesterol into the cell membrane can rescue the decrease of phospho-Akt induced by MβCD treatment in *CG5399*-overexpressing cells, excluding the possibility that an off-target effect of MβCD contributes to the observed effect (*Figure 4—figure supplement 1C*). To test the possibility that MβCD treatment affected the normal function of InR, S2*R*+ cells were stimulated with insulin following MβCD treatment. No difference in insulin response was observed in MβCD treated and untreated cells (*Figure 4—figure supplement 1D*), suggesting that MβCD treatment does not affect InR function. Moreover, direct supplementation of cholesterol to the cell membrane activated InR-Akt-mTOR signaling in wild-type S2*R*+ cells (*Figure 4B*), indicating that an increase in the level of cholesterol at the cell membrane was able to activate InR.

Cell membranes form distinct microdomains, such as caveolin-coated caveolae, flotillin-coated microdomains, and clathrin-coated pits. Cholesterol is required for the formation of different microdomains (*Lu and Fairn, 2018*). To distinguish which microdomain was involved in the *CG5399* function, we knocked down flotillins and clathrins in *CG5399*-overexpressing cells using dsRNAs. Knockdown of flotillin genes (*Flo1* and *Flo2*) did not affect phospho-Akt, whereas the increase of phospho-InR and phospho-Akt normally observed in *CG5399*-overexpression cells was dampened by knockdown of clathrin heavy chain (*Chc*) (*Figure 4C and D*). Moreover, InR activation by cholesterol supplementation was also eliminated by *Chc* knockdown (*Figure 4E*). Taken together, these results suggest that InR-Akt-mTOR signaling activation by *CG5399* overexpression requires cholesterol and clathrin-coated pits at the cell membrane.

## Discussion

Although genome-wide LOF screens in *Drosophila* cells have helped elucidate the mechanism of a variety of biological processes, genome-wide GOF screens have not been feasible in this organism. To address this gap, we generated a genome-wide dual-sgRNA library that covers both protein-coding

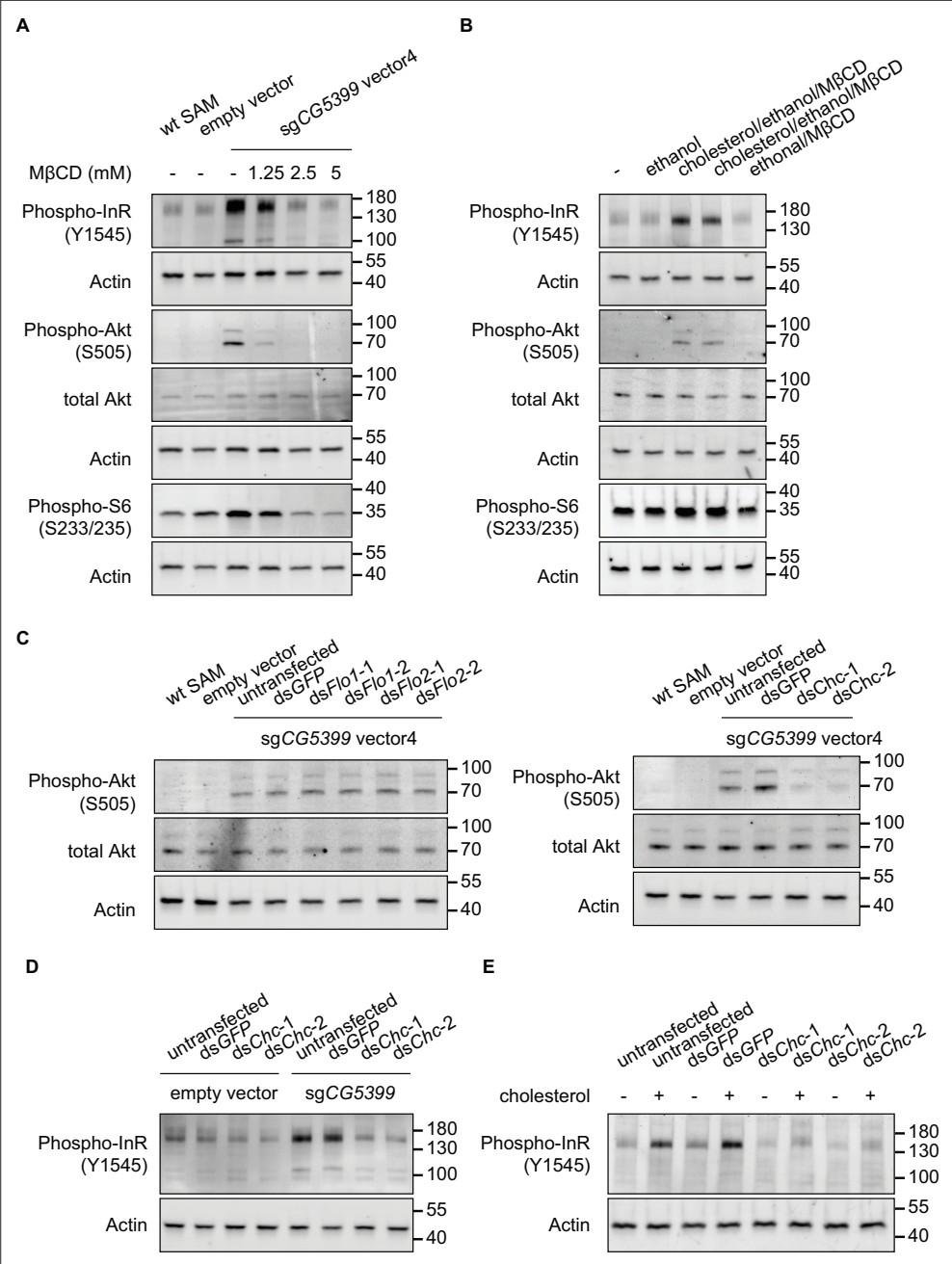

**Figure 4.** Activation of InR-Akt-mTOR signaling by *CG5399* overexpression requires cholesterol and clathrin-coated pits at the membrane. (**A**) Phospho-InR, phospho-Akt, and phospho-S6 in *CG5399*-overexpressing cells treated with methyl-beta-cyclodextrin (MβCD) at different concentrations. (**B**) Phospho-InR, phospho-Akt, and phospho-S6 in S2*R*+ cells with cholesterol supplementation. Two different cholesterol products from Sigma (C3045 for Lane 3 and C2044 for Lane 4) were used. (**C**) Phospho-Akt in *CG5399*-overexpressing cells following *Flo1*, *Flo2,* or *Chc* knockdown. Two nonoverlapping double-stranded RNAs (dsRNAs) targeting each gene were used. (**D**) Phospho-InR in *CG5399*-overexpressing cells following *Chc* knockdown. Two nonoverlapping dsRNAs targeting each gene were used. (**E**) Phospho-InR in S2*R*+ cells with cholesterol supplementation following clathrin heavy chain (*Chc*) knockdown. Two nonoverlapping dsRNAs targeting each gene were used.

The online version of this article includes the following source data and figure supplement(s) for figure 4:

**Source data 1.** Full source data for *Figure 4*.

**Figure supplement 1.** *CG5399* interacts with cholesterol by molecular docking.

**Figure supplement 1—source data 1.** Full source data for *Figure 4—figure supplement 1*.

genes and non-coding genes. This library also captures transcriptional complexity by targeting alternative promoters of the same gene, making it suitable for screens in different contexts in the future. Moreover, in this library, each gene is targeted by 4–6 dual-sgRNA vectors, which helps overcome the inefficient activation by some sgRNAs to some extent. Indeed, we observed that some dual-sgRNA vectors failed to activate the target genes (*Figure 2F and I*), possibly due to steric hindrance by pre-existing proteins in the promoter regions, or due to different sgRNA binding efficiency, which might be optimized by applying machine learning to large screen datasets in the future. In our method, the synthetic transcriptional activators of the SAM complex (dCas9-VP64 and MCP-p65-HSF1) are driven by the metallothionein promoter. Due to the leaky expression of this promoter, we observed moderate gene activation without copper induction. Availability of a tighter controlled promoter may be required for more sensitive screens. As the *Drosophila* genome is relatively compact, one concern for CRISPRa is the collateral activation of adjacent genes. Previous work has shown that sgRNAs targeting areas beyond –600 bp from the TSS in *Drosophila* lose efficiency (*Mao et al., 2020*), which may alleviate the concern of collateral activation when the sgRNAs target sites are over 600 bp away from the TSS of neighboring genes. In some cases, sgRNAs are designed within both promoters of closely spaced divergent genes. If the hits from the screen form closely spaced divergent gene pairs, e.g., *Lis-1* and *Ptp52F* in the cell fitness screen (*Table 1*), more experiments will be required to identify which one contributes to the phenotype. Currently, in our CRISPRa system, spCas9 requires NGG as a protospacer-adjacent motif (PAM) sequence, which limits sgRNA design in the small region upstream of TSS, especially for closely spaced genes. Next-generation CRISPRa system could use PAMless Cas9 variants (*Walton et al., 2020*) to remove the PAM constraint for sgRNA design, allowing the construction of libraries with more sgRNAs per gene.

Our genome-wide genetic screen identified some known rapamycin resistance genes and novel candidates. Overexpression of *CG5399*, which encodes a lipocalin family protein, confers resistance to rapamycin and activates RTK-Akt-mTOR signaling. The activation of InR by *CG5399* requires cholesterol and clathrin-coated pits at the cell membrane (*Figure 4—figure supplement 1E*). CG5399 is predicted to have a transmembrane helix at the C-terminus by PredictProtein and be located on the cell membrane by DeepLoc. As InR is embedded in the cell membrane and the rearrangement of InR transmembrane domain is crucial for tyrosine kinase domain activation, changing the lipid environment is a reasonable possible mechanism for the regulation of InR activation (*Gutmann et al., 2018*; *Scapin et al., 2018*). A recent study showed that lipid exchange to form ordered domains at cell membranes induces InR autophosphorylation (*Suresh et al., 2021*). In our experiment, we observed higher InR activity in *CG5399*-overexpressing cells without insulin stimulation, suggesting that InR activation can be regulated by the lipid environment. The clinical relevance of this InR activation mechanism needs to be further investigated.

In conclusion, we have established a genome-wide CRISPRa platform in *Drosophila* cells and identified novel rapamycin resistance genes using a genome-wide CRISPRa screen platform. This platform can be applied broadly to help elucidate the cellular mechanisms of a variety of biological processes.

## Materials and methods

**Key resources table**

| Reagent type (species) or resource | Designation | Source or reference | Identifiers | Additional information |
|---|---|---|---|---|
| Gene (*Drosophila melanogaster*) | CG8468 | FlyBase | FLYB:FBgn0033913 | |
| Gene (*Drosophila melanogaster*) | CG5399 | FlyBase | FLYB:FBgn0038353 | |
| Gene (*Drosophila melanogaster*) | CG9932 | FlyBase | FLYB:FBgn0262160 | |
| Cell line (*D. melanogaster*) | S2R+ | DRSC | FLYB:FBtc0000150 | |
| Cell line (*D. melanogaster*) | PT5 | DRSC | FLYB:FBtc0000229 | |
| Strain, strain background (*Escherichia coli*) | E.cloni10GF' Electrocompetent Cells | Biosearch Technologies | 60061–2 | sgRNA library construction |

*Continued on next page*

*Continued*

| Reagent type (species) or resource | Designation | Source or reference | Identifiers | Additional information |
|---|---|---|---|---|
| Strain, strain background (*Escherichia coli*) | One Shot TOP10 Chemically Competent *E. coli* | Invitrogen | C404010 | |
| Antibody | Recombinant Anti-Insulin Receptor (phospho Y1185) antibody (Rabbit monoclonal) | Abcam | ab62321 | 1:1000 for WB |
| Antibody | Phospho-Akt (Ser473) (D9E) XP antibody (Rabbit monoclonal) | Cell Signaling Technology | 4060 | 1:1000 for WB |
| Antibody | Akt Rabbit Antibody (Rabbit polyclonal) | Cell Signaling Technology | 9272 | 1:1000 for WB |
| Antibody | StarBright Blue 700 Goat Anti-Rabbit IgG | Bio-Rad | 12004161 | 1:2500 for WB |
| Antibody | StarBright Blue 520 Goat Anti-Rabbit IgG | Bio-Rad | 12005869 | 1:2500 for WB |
| Antibody | hFAB Rhodamine Anti-Actin Primary Antibody (synthesized, monoclonal) | Bio-Rad | 12004163 | 1:2500 for WB |
| Recombinant DNA reagent | pMK33-SAM plasmid | This paper | | Can be obtained from DRSC |
| Recombinant DNA reagent | pLib8 plasmid | This paper | | U6:3-MS2 sgRNA cassette, can be obtained from DRSC |
| Recombinant DNA reagent | pBS130 plasmid | Addgene | 26290 | PhiC31 integrase |
| Recombinant DNA reagent | pUAS-CG5399 plasmid | This paper | | CG5399 ORF vector, cassette, can be obtained from DRSC |
| Commercial assay or kit | Effectene Transfection Reagent | Qiagen | 301425 | |
| Commercial assay or kit | CellTiter-Glo Luminescent Cell Viability Assay | Promega | G7570 | |
| Commercial assay or kit | RNeasy Mini Kit | Qiagen | 74104 | |
| Commercial assay or kit | iScript cDNA Synthesis Kit | Bio-Rad | 1708890 | |
| Chemical compound, drug | MEGAscript T7 Transcription Kit | Invitrogen | AM1334 | |
| Chemical compound, drug | Methyl-β-cyclodextrin | Sigma-Aldrich | C4555 | |
| Chemical compound, drug | Cholesterol | Sigma-Aldrich | C3045 | |
| Chemical compound, drug | Cholesterol | Sigma-Aldrich | C2044 | |
| Software, algorithm | GraphPad Prism 7 | GraphPad | | |
| Software, algorithm | FlowJo | FlowJo | | |

## Vectors

The pMK33-SAM plasmid was generated by transferring the SAM sequence from the flySAM vector (*Jia et al., 2018*) into the pMK33 plasmid. MS2 hairpin containing sgRNA was expressed from the pLib8 plasmid. pLib8 is derived from pLib6.4 (*Viswanatha et al., 2018*) by replacing the U6:2-sgRNA cassette to the U6:3-MS2 sgRNA cassette. The PhiC31 integrase expressing pBS130 plasmid was obtained from Addgene (#26290). The full length of *CG5399* was cloned from the cDNA of S2R+ cells and inserted into the pWalium10 vector (DGRC, 1470) to construct the *pUAS-CG5399* vector. sgRNA sequences used in this study are listed in *Supplementary file 1*.

## Antibodies

Phospho-InR antibody (Abcam, #ab62321), Phospho-Akt antibody (Cell signaling, # 4060), Akt antibody (Cell signaling, #9272) were used in this study. Phospho-S6 antibody is a kind gift of *Kim and Choi, 2019*.

## Cell culture, transfection, and proliferation assay

*Drosophila* cells were cultured with Schneider medium (Gibco) supplemented with 10% heat-inactivated FBS (Gibco) at 25°C (*Viswanatha et al., 2018*) unless otherwise indicated. The

wild-type S2R+ cell line and the *attP* sites containing S2R+ derivative PT5 cell line were obtained from the *Drosophila* RNAi Screening Center. The DRSC copy of S2R+ was authenticated by the *Drosophila* Genomics Resource Center (DGRC), as part of their development of a transposable element-based authentication protocol for *Drosophila* cell lines (*Mariyappa et al., 2022*). The presence of the recombination cassette was confirmed by observation of mCherry fluorescence and the successful introduction of sgRNAs via recombination-mediated cassette exchange. Mycoplasma contamination is not an issue for *Drosophila* cell lines; throughout the study, we monitored and confirmed through careful observation that media and cells were not infected by bacteria or fungi. The cell lines used in this study are not from the list of commonly misidentified cell lines maintained by the International Cell Line Authentication Committee. PT5 cells were transfected with pMK33-SAM plasmid using Effectene (QIAGEN) according to the manufacturer's protocol. Briefly, $3 \times 10^6$ *Drosophila* cells were seeded into one well of a six-well plate before transfection. 400 ng plasmids were diluted into Buffer EC to a final volume of 100 μL and mixed with 3.2 μL enhancer by vortexing to form a DNA-enhancer mixture. 10 μL Effectene transfection reagents were added into the DNA-enhancer mixture and mixed by vortexing. After incubating at room temperature for 15 min to allow transfection complex formation, the solution was added drop-wise onto *Drosophila* cells. Transfected cells were passaged in a culture medium containing 200 μg/mL Hygromycin B (Millipore) for 1 month to generate the stable SAM cell line. To induce SAM complex expression, a culture medium containing 100 μM $CuSO_4$ (Sigma) was used. Cell proliferation under different rapamycin concentrations was tested using CellTiter Glo assay (Promega) according to the manufacturer's protocol. $1 \times 10^4$ *Drosophila* cells were seeded into each well of a 96-well plate. Rapamycin-containing culture medium was added into each well to make the final concentration from $10^{-4}$ nM to 10 nM. After culturing for 4 days, a volume of CellTiter Glo reagent was added into each well before cells reached confluence. The luminescence signal was measured by Plate Reader (Molecular Devices).

## Pooled library design and construction

For the focused sgRNA library, sgRNAs were designed within 500 bp upstream of the transcriptional start site (TSS) for each gene. Ten different sgRNAs were selected for each gene unless fewer sgRNA binding sites were found within the window. Constructing the focused library was performed as previously described (*Viswanatha et al., 2018*). Briefly, Bbs1 sites flanking sgRNA spacer sequences were synthesized as single-stranded DNA oligos (Agilent). DNA oligos were amplified by PCR using Phusion Polymerase (New England Biolabs). Bbs1 restriction enzyme (New England Biolabs) was used to digest amplicon and pLib8 plasmid. The resulting 24-mer fragment was purified from the Bbs1 digested amplicon by running a 20% TBE polyacrylamide gel (Thermo). The purified fragment and plasmid were ligated using T4 ligase (New England Biolabs). The ligation products were transferred into Ecloni 10GF' electrocompetent cells (Lucigen) using Gene Pulser Xcell Electroporation Systems (Bio-Rad). Transformed bacteria were spread on LB-carbenicillin agar plates. After overnight culture, the bacteria colonies were collected from plates by scraping and amplified in an LB medium with ampicillin.

For the genome-wide dual-sgRNA library, sgRNAs were designed within 500 bp upstream of TSS for each gene. sgRNAs were chosen to make six dual-sgRNA combinations for each gene unless fewer sgRNA binding sites were found. To construct the dual-sgRNA library, Bbs1, and BsmB1 sites flanking two sgRNA spacer sequences were synthesized in a custom array as single-stranded DNA oligos (Agilent). DNA oligos were amplified by PCR using Phusion Polymerase (New England Biolabs). DNA amplicons were ligated to Zero Blunt vector (Thermo) using T4 ligase to generate the first library. An amplicon containing the scaffold sequence for the first sgRNA and U6:2 promoter sequence for the second sgRNA was inserted into the BsmB1 site to generate the second library. The second library and pLib8 vector were digested with Bbs1. The resulting sgRNA cassettes from the digested second library were ligated with Bbs1 digested pLib8 vector to generate the final library using T4 ligase. Each library was transferred into Ecloni 10GF' electrocompetent cells by electroporation. Transformed bacteria were spread on LB-carbenicillin agar plates. Bacteria colonies were calculated by serial dilution. Each library needs to reach at least 10 times diversity to maintain the integrity of the library.

## Pooled screening, library sequencing, and data analysis

The library was co-transfected with the same amount of phiC31 plasmid into SAM cells at $3 \times 10^6$ cells/well of a six-well plate using Effectene. The total cell number used for transfection was calculated to ensure over 1000 cells/sgRNA to maintain the integrity of the library. The transfected cells were passaged in a culture medium containing 5 μg/mL puromycin for 3 weeks to select the sgRNA-integrated cells. The resulting pooled library cells were split into two populations and passaged in rapamycin or DMSO-containing medium supplemented with 100 μM $CuSO_4$ for indicated days. After treatment, the genomic DNA was extracted from the final cell population and the sgRNAs sequences were amplified by PCR. As each *Drosophila* cell contains ~0.6 pg DNA, the amount of genomic DNA used as a PCR template was calculated to ensure over 1000 cells/sgRNA to maintain the diversity. The library for next-generation sequencing was constructed by adding Illumina adaptors to sgRNA amplicons by PCR. The final PCR products had the following sequence: P5-read1-$(N)_n$-$(B)_6$-sgRNA-P7 (where N stands for any nucleotide, n stands for a variable length of nucleotide from 1 to 10, and $(B)_6$ stands for six nucleotides sample barcode). PCR primers used for NGS library construction are listed in *Supplementary file 2*. The library was sequenced using the NextSeq500 $1 \times 75$ SE platform (Illumina) in HMS Biopolymers Facility. The sequencing data were de-multiplexed using TagDust. Screen hits were identified using MAGeCK-RRA by comparing the treatment and the control according to the previous report (*Li et al., 2014*).

## RNA extraction, reverse transcription, and qPCR

RNA was extracted from *Drosophila* cells using RNA Mini Kit (QIAGEN) according to the manufacturer's protocol. Total RNA was reverse transcribed into cDNA using the iScript cDNA Synthesis Kit (Bio-Rad). qPCR was done with SYBR Green Master Mix (Bio-Rad). The housekeeping gene *rp49* was used as the reference gene for qPCR. qPCR primers used in this study are listed in *Supplementary file 2*. The statistical analysis was performed using the GraphPad Prism 7 software. *t*-tests were performed to test the significance of gene expression data. *p<0.05; **p<0.01; ***p<0.001; ns, not significant.

## GFP proportion analysis by flow cytometry

Cells were transferred into 5 mL FACS tubes (Falcon 352235) and analyzed with a BD LSR II Flow Cytometer in the Department of Immunology Flow Cytometry Facility, Harvard Medical School. Wild-type cells (GFP negative cells) and an empty vector expressing cells (GFP positive cells) were used as a negative and positive control to set the gate in Alexa Fluor 488 channel, respectively. With this gate, the GFP proportion in wild-type cells is 0.031%, and in an empty vector expressing cells is 96.7%. Three biological replicates for each condition were tested. The flow cytometry files were analyzed by FlowJo. The statistical analysis was performed using the GraphPad Prism 7 software. *t*-tests were performed to test the significance of GFP proportion data. *p<0.05; **p<0.01; ***p<0.001; ns, not significant.

## dsRNA synthesis and transfection

dsRNAs were designed by the *Drosophila* RNAi Screening Center. dsRNA templates were amplified from genomic DNA using primers with T7 promoter sequence TAATACGACTCACTATAGGG at 5' end. dsRNAs were synthesized from the resulting amplicons using MEGAscript T7 Transcription Kit (Invitrogen). dsRNAs were purified with RNeasy Mini Kit (QIAGEN) before transfection. dsRNA sequences used in this study are listed in *Supplementary file 3*. 10 μg dsRNA were transfected into $3 \times 10^6$ *Drosophila* cells using Effectene (QIAGEN).

## Cholesterol depletion and supplementation

For cholesterol depletion, methyl-beta-cyclodextrin (sigma) was dissolved in serum-free Schneider medium at the indicated concentration. Cells were incubated with MβCD containing serum-free medium for 1 hr before testing. For cholesterol supplementation, cholesterol (sigma) was dissolved in ethanol. Dissolved cholesterol was added into MβCD containing serum-free Schneider medium to form cholesterol/MβCD complex. Cells were incubated with cholesterol/MβCD complex containing serum-free Schneider medium for 1 hr before testing.

## Acknowledgements

We thank Dr. Jianquan Ni for the flySAM vector and Dr. Ah-Ram Kim for the phospho-S6 antibody. This work is supported by NIH NIGMS P41 GM132087 (NP) and NIH P01CA120964. NP is an investigator at Howard Hughes Medical Institute.

## Additional information

### Funding

| Funder | Grant reference number | Author |
| --- | --- | --- |
| National Institute of General Medical Sciences | GM132087 | Stephanie E Mohr Norbert Perrimon |
| National Cancer Institute | CA120964 | Norbert Perrimon |
| Howard Hughes Medical Institute | | Norbert Perrimon |
| NIH | P01CA120964 | Norbert Perrimon |

The funders had no role in study design, data collection and interpretation, or the decision to submit the work for publication.

### Author contributions

Baolong Xia, Conceptualization, Formal analysis, Investigation, Writing - original draft, Project administration; Raghuvir Viswanatha, Yanhui Hu, Formal analysis; Stephanie E Mohr, Funding acquisition, Writing – review and editing; Norbert Perrimon, Conceptualization, Supervision, Funding acquisition, Writing – review and editing

### Author ORCIDs

Baolong Xia http://orcid.org/0000-0003-2536-0267
Raghuvir Viswanatha http://orcid.org/0000-0002-9457-6953
Stephanie E Mohr http://orcid.org/0000-0001-9639-7708
Norbert Perrimon http://orcid.org/0000-0001-7542-472X

### Decision letter and Author response

Decision letter https://doi.org/10.7554/eLife.85542.sa1
Author response https://doi.org/10.7554/eLife.85542.sa2

## Additional files

### Supplementary files
- Supplementary file 1. sgRNA vectors used in this study.
- Supplementary file 2. PCR primers used in this study.
- Supplementary file 3. dsRNAs used in this study.
- MDAR checklist
- Source data 1. The original files of the full raw unedited blots and figures with the uncropped blots with relevant bands labeled in this study.

### Data availability

All data generated or analysed during this study are included in the manuscript and source data files. The pMK33-SAM vector, pLib8 vector, and libraries used in this study are available through DRSC/TRiP Functional Genomics Resources.

The following dataset was generated:

| Author(s) | Year | Dataset title | Dataset URL | Database and Identifier |
|---|---|---|---|---|
| Xia B, Viswanatha R, Hu Y, Mohr SE, Perrimon N | 2023 | Data from: Pooled genome-wide CRISPR activation screening for rapamycin resistance genes in *Drosophila* cells | https://dx.doi.org/10.5061/dryad.2547d7ww8 | Dryad Digital Repository, 10.5061/dryad.2547d7ww8 |

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
