## [Editor Report]

This work is well-structured, with clear objectives and experiments. The authors successfully demonstrated genome-wide gene activation which overcame previous failed attempts to replicate gene activation that worked well in mammalian systems. The study is detailed and relevant for the application of CRISPRa in understanding the function of candidate genes.

---

## [Decision Letter]

**Decision letter after peer review:**

Thank you for submitting your article "Pooled genome-wide CRISPR activation screening for rapamycin resistance genes in *Drosophila* cells" for consideration by *eLife*. Your article has been reviewed by 2 peer reviewers, and the evaluation has been overseen by a Reviewing Editor and K VijayRaghavan as the Senior Editor. The following individual involved in the review of your submission has agreed to reveal their identity: Raghu Padinjat (Reviewer #2).

The reviewers have discussed their reviews with one another, and the Reviewing Editor has drafted this to help you prepare a revised submission. As you can see, the suggestions for revision are readily addressable. We look forward to receiving a revised manuscript speedily.

Essential revisions:

While we strongly recommend this manuscript for publication some corrections and revisions are listed.

1. The authors used attP cells to integrate sgRNA, and it would be useful to know the copy number of attP sites per cell and whether the cells are homogeneous. If this information is available, it should be added to the manuscript.

2. Due to the inherent nature of the metallothionein promoter, the authors observed significant leaky activation. While this may not be an issue while studying the rapamycin pathway, it could be problematic when studying sensitive signaling processes. Therefore, a clear disclaimer should be added to the discussion.

3. The manuscript demonstrates that CRISPRa is specific and does not activate neighboring genes with just a single example. However, it is possible that insulator sequences protect neighboring gene activation in this case. Therefore, the authors should either show more examples to demonstrate this convincingly or explicitly state this in the discussion.

4. It would be helpful to know the abundance of gRNA in the library compared to the hits identified in the screen. Was there a greater probability of a target showing up as positive in the screen when its abundance is overrepresented in the library?

5. The authors may want to speculate as to why Cyp12a4 did not come up on the second screen.

6. It would be useful to know whether p-Akt was checked in CG9932 knockdown and why it was excluded from the experiment.

7. The experiments to suggest insulin-independent activation of InR by CG5399 require more conclusive experiments. Serum starvation is not sufficient to conclusively state this, and the authors should discuss this limitation.

8. The discussion lacks clarity and focus, particularly in the second paragraph. Instead, the authors could focus on the tools generated in the article and compare and discuss similar tools available in other model systems. They could also speculate on how such a tool in a *Drosophila* model will serve in new research questions.

9. Although the work seems more focused on generating a fly tool for CRISPRa, there are some nice discoveries of new functions for some of the fly genes. This could be better illustrated with a model showing signaling pathways and highlighting the gene functions identified in this study. It would be helpful to include an overall description of the InK-AktmTOR pathway with only major signaling cascade molecules for understanding.

10. It is recommended to expand abbreviations at their first mention. For example, the full form of InR should be included in Line 51.

11. It would be helpful to briefly explain the role of each component of the SAM complex- VP64, MS2, p65, and HSF1 for a better understanding of the use of the SAM complex.

12. In Line 100 and in a few parts of the same section, it is unclear what "cell fitness" indicates. It would be useful to specify whether it refers to cell viability, proliferation rate, or cell functionality and what parameters were measured to assess fitness.

13. In Line 124, the authors should state the rationale for including non-coding genes in the study with respect to rapamycin resistance. Were any non-coding targets identified in the screen?

14. The details of the software used for statistical analysis are missing. The details of FACS, its gating, and control parameters could be added.

15. In general, the legends for the figures could be better explained in detail. Eg: Appropriate statistical explanations for * needed wherever mentioned in figures. Abbreviations in plasmid constructs needed better explanations.

16. Authors have taken a number of measures to maintain the integrity of the CRISPRa library, including multiple gRNA targets per gene, 1000 cells per gRNA, and deep sequencing. However, do the authors have an idea of what percentage of the gRNA vectors are functional? Looking at the data they show for the 3 candidate genes, at least half of them are not functional, which could be either because of gRNA location or efficiency. Considering this to be an average situation, there might be a large number of genes for which all gRNAs might not function at all. We understand this might be a caveat for all such studies, but an estimate of some kind in discussion might be useful for anyone who might want to use these resources.

17. As the authors mention that ~32% of genes in *Drosophila* have transcription start sites <1kb apart, off-targeting (neighboring genes getting activated in addition to the intended gene) will be an issue. To address this, the authors describe one example of genes where although the genes TSS are within one kb of each other, the sgRNA specifically activated only one gene and not the other. However, since following this, authors have generated genome-wide resources keeping 500bp upstream as their benchmark, a large percentage of these 32% genes might have off-targets. It would be useful to know the estimates of off-targeting for such a resource. In addition, have authors looked at the transcripts of genes close to the specific genes they have studied? CG9932 is in close proximity to (although not within the 1 kb range) a few genes including mTOR.

---

## [Author Response]

Essential revisions:While we strongly recommend this manuscript for publication some corrections and revisions are listed.1. The authors used attP cells to integrate sgRNA, and it would be useful to know the copy number of attP sites per cell and whether the cells are homogeneous. If this information is available, it should be added to the manuscript.

The *attP* sites containing S2R+ derivative PT5 cell line used in this study has been described in our previous publication (PMID: 22174071). The *attP* cassette was inserted into the genome by MiMIC transposition. The PT5 cell line was established from a single cell-derived clone after MiMIC transposon insertion and the insertion site was mapped to the *Clic* locus by inverse PCR. We have added this information to the manuscript.

2. Due to the inherent nature of the metallothionein promoter, the authors observed significant leaky activation. While this may not be an issue while studying the rapamycin pathway, it could be problematic when studying sensitive signaling processes. Therefore, a clear disclaimer should be added to the discussion.

We agree that the leaky expression from the metallothionein promoter might be an issue for some applications. We have added the following text to the discussion: “In our method, the synthetic transcriptional activators of the SAM complex (dCas9-VP64 and MCP-p65-HSF1) are driven by the metallothionein promoter. Due to the leaky expression of this promoter, we observed moderate gene activation without copper induction. Availability of a tighter controlled promoter may be required for more sensitive screens.”

3. The manuscript demonstrates that CRISPRa is specific and does not activate neighboring genes with just a single example. However, it is possible that insulator sequences protect neighboring gene activation in this case. Therefore, the authors should either show more examples to demonstrate this convincingly or explicitly state this in the discussion.

We did not systematically analyze the collateral activation of neighboring genes in our study. However, previous work from others (PMID: 33020192) have shown that sgRNAs targeting areas beyond -600 bp from the TSS lose efficiency in *Drosophila*, which may alleviate the concern of collateral activation of neighboring genes when the sgRNAs are over 600 bp away. We have added the following text to the discussion: “As the *Drosophila* genome is relatively compact, one concern for CRIPRa is collateral activation of adjacent genes. Previous work has shown that sgRNAs targeting areas beyond -600 bp from the TSS in *Drosophila* lose efficiency (PMID: 33020192), which may alleviate the concern of collateral activation when the sgRNAs target sites are over 600 bp away from the TSS of neighboring genes.”

4. It would be helpful to know the abundance of gRNA in the library compared to the hits identified in the screen. Was there a greater probability of a target showing up as positive in the screen when its abundance is overrepresented in the library?

That is not the case. We ranked sgRNAs based on their abundance in the initial population and highlighted sgRNAs of the top three hits from the rapamycin screen. The sgRNAs of the top hits are not overrepresented in the initial population. In fact, the hits are identified by the change of the sgRNAs in the Rapamycin treated population compared to DMSO treated population. It is not related to the sgRNA abundance in the initial population.

**Author response image 1. sa2fig1:** Reads of *CG5399* sgRNAs in the initial population (*CG5399* sgRNAs are in red, other sgRNAs are in gray).

**Author response image 2. sa2fig2:** Reads of *CG8468* sgRNAs in the initial population (*CG8468* sgRNAs are in red, other sgRNAs are in gray).

**Author response image 3. sa2fig3:** Reads of *CG9932* sgRNAs in the initial population (*CG9932* sgRNAs are in red, other sgRNAs are in gray).

5. The authors may want to speculate as to why Cyp12a4 did not come up on the second screen.

*Cyp12a4* was identified as a hit affecting cell fitness from the screen with the focused library but was not identified from the screen with the genome-wide library. There are ~10 sgRNA vectors per gene in the focused library while ~ 6 sgRNA vectors per gene in the genome-wide library. We speculate that the sgRNA vector number per gene in different libraries might affect the power to identify some hits. It suggests that a next generation CRISPRa library should include more sgRNA vectors per gene.

6. It would be useful to know whether p-Akt was checked in CG9932 knockdown and why it was excluded from the experiment.

Previous work has shown that overexpression of *REST*, the human ortholog of *CG9932*, activates Akt (PMID: 30670636). As it is consistent with what we observed in fly cells, we did not follow up with the characterization of *CG9932*.

7. The experiments to suggest insulin-independent activation of InR by CG5399 require more conclusive experiments. Serum starvation is not sufficient to conclusively state this, and the authors should discuss this limitation.

In this study, we observed InR activation by *CG5399* overexpression without insulin treatment, and all the experiments were done without insulin stimulation. *Drosophila* cells are cultured in Schneider medium supplemented with 10% FBS (fetal bovine serum). The Schneider medium only consists of amino acids and inorganic salts while FBS is a biological product which might contain trace amount of insulin and insulin-like growth factors (PMID: 30014700). To further exclude the effect of insulin and insulin-like growth factors in FBS, we removed FBS from the culture medium. We have added this information to the manuscript to clarify the design of the serum starvation experiment.

8. The discussion lacks clarity and focus, particularly in the second paragraph. Instead, the authors could focus on the tools generated in the article and compare and discuss similar tools available in other model systems. They could also speculate on how such a tool in a *Drosophila* model will serve in new research questions.

We have modified our discussion. We deleted the original second paragraph and added a discussion on the limitations of our current CRISPRa system and the design of next generation CRISPRa system.

9. Although the work seems more focused on generating a fly tool for CRISPRa, there are some nice discoveries of new functions for some of the fly genes. This could be better illustrated with a model showing signaling pathways and highlighting the gene functions identified in this study. It would be helpful to include an overall description of the InK-AktmTOR pathway with only major signaling cascade molecules for understanding.

We have added a schematic model of InR-Akt-mTOR activation in *CG5399* overexpressing cells to Figure 4—figure supplement 1E.

10. It is recommended to expand abbreviations at their first mention. For example, the full form of InR should be included in Line 51.

We have added the full name of RTK in Line 49 and the full name of InR in the abstract.

11. It would be helpful to briefly explain the role of each component of the SAM complex- VP64, MS2, p65, and HSF1 for a better understanding of the use of the SAM complex.

We have added the following text to the manuscript to describe the role of each component of the SAM complex: “VP64, p65 and HSF1 are transcriptional activators while the MS2 coat protein recognizes and binds to MS2 hairpins present in the sgRNAs, recruiting the fused transcriptional activators to gene promoters targeted by sgRNAs.”

12. In Line 100 and in a few parts of the same section, it is unclear what "cell fitness" indicates. It would be useful to specify whether it refers to cell viability, proliferation rate, or cell functionality and what parameters were measured to assess fitness.

As the readout is the sgRNA abundance in the final population, cell fitness in this study is a combined effect from cell viability, proliferation rate and cell functionality.

13. In Line 124, the authors should state the rationale for including non-coding genes in the study with respect to rapamycin resistance. Were any non-coding targets identified in the screen?

As we wanted to establish a universal platform for the *Drosophila* community, the genome wide library design also includes non-coding genes for a broader application. One non-coding gene (*CR44587*) was identified as a hit affecting cell fitness (Table 1) and another non-coding gene (*CR43417*) was identified as a hit conferring rapamycin resistance in one replicate of the screen (Figure 2B).

14. The details of the software used for statistical analysis are missing. The details of FACS, its gating, and control parameters could be added.

For the software used for statistical analysis, we have added the following text to the Materials and methods: “The statistical analysis was perfomed using the GraphPad Prism 7 software.” For the FACS and gating, we have added the following text to the Materials and methods: “Cells were transferred into 5 mL FACS tubes (Falcon 352235) and analyzed with a BD LSR II Flow Cytometer in the Department of Immunology Flow Cytometry Facility, Harvard Medical School. Wildtype cells (GFP negative cells) and empty vector expressing cells (GFP positive cells) were used as negative and positive control to set the gate in Alexa Fluor 488 channel, respectively. With this gate, the GFP proportion in wildtype cells is 0.031% and in empty vector expressing cells is 96.7%. Three biological replicates for each condition were tested. The flow cytometry files were analyzed by Flowjo. The statistical analysis was performed using the GraphPad Prism 7 software. t-tests were performed to test the significance of GFP proportion data. *P < 0.05; **P < 0.01; ***P < 0.001; ns, not significant.”

15. In general, the legends for the figures could be better explained in detail. Eg: Appropriate statistical explanations for * needed wherever mentioned in figures. Abbreviations in plasmid constructs needed better explanations.

We have added the statistical analysis used in each figure. We have added the following text to the figure 1A legend: “dCas9-VP64 and MCP-p65-HSF1 were expressed as T2A-containing bicistronic transcript. sgRNA was expressed from pLib8 plasmid, which contains a *attB* flanking GFP-T2A-PuroR cassette for *attP* sites recombination.”

16. Authors have taken a number of measures to maintain the integrity of the CRISPRa library, including multiple gRNA targets per gene, 1000 cells per gRNA, and deep sequencing. However, do the authors have an idea of what percentage of the gRNA vectors are functional? Looking at the data they show for the 3 candidate genes, at least half of them are not functional, which could be either because of gRNA location or efficiency. Considering this to be an average situation, there might be a large number of genes for which all gRNAs might not function at all. We understand this might be a caveat for all such studies, but an estimate of some kind in discussion might be useful for anyone who might want to use these resources.

Previous CRISPRa studies have shown that activation efficiency is affected by the basal expression level and epigenetic state of target genes (PIMD: 25494202, 36917981). Moreover, for the same genes, we observed that some sgRNA vectors can activate target genes while others cannot (Figure 2f and 2I). In our original manuscript, we speculated that this is possibly due to steric hindrance by pre-existing proteins in the promoter regions, or due to different sgRNA binding efficiency, which can be further optimized in the future by applying machine learning to large screen datasets. Although we do not know how many sgRNAs are functional in the library, it is worth noting that while some sgRNA vectors fail to activate CG5399 and CG9932, these two genes can still be identified as top hits by MAGeCK-RRA analysis. Next generation CRISPRa sgRNA library should include more sgRNA vectors per gene to overcome the inefficient activation of some sgRNAs. We have added the following text to the discussion: “Currently in our CRISPRa system, spCas9 requires NGG as protospacer-adjacent motif (PAM) sequence, which limits sgRNA design in the small region upstream of TSS, especially for closely spaced genes. Next generation CRISPRa system could use PAMless Cas9 variants to remove the PAM constraint for sgRNA design, allowing the construction of libraries with more sgRNAs per gene.”

17. As the authors mention that ~32% of genes in *Drosophila* have transcription start sites <1kb apart, off-targeting (neighboring genes getting activated in addition to the intended gene) will be an issue. To address this, the authors describe one example of genes where although the genes TSS are within one kb of each other, the sgRNA specifically activated only one gene and not the other. However, since following this, authors have generated genome-wide resources keeping 500bp upstream as their benchmark, a large percentage of these 32% genes might have off-targets. It would be useful to know the estimates of off-targeting for such a resource. In addition, have authors looked at the transcripts of genes close to the specific genes they have studied? CG9932 is in close proximity to (although not within the 1 kb range) a few genes including mTOR.

When we designed the genome-wide sgRNA library, if a sgRNA is located within 500 bp upstream of two genes, both genes are annotated as the target genes of the sgRNA. In our genome-wide sgRNAs, 14% of sgRNA vectors are annotated to target more than one gene. In the original manuscript, we discussed that if the hits from the screen form closely spaced divergent gene pairs, more experiments will be required to identify which one contributes to the phenotype. We have also added some discussion to the manuscript about next generation sgRNA library with closer space to target gene using PAMless Cas9 variants. For *CG8468*, *CG5399* and *CG9932*, none of them form divergent gene pairs with neighboring genes within 1 kb. *CG9932* is over 50 kb away from mTOR, sgRNAs targeting *CG9932* are unlikely to active mTOR according to the previous study (PMID: 33020192). Moreover, for *CG5399*, we have excluded the off-target effect of sgRNAs by dsRNA and ORF overexpression.